# Dynamics and Determinants of SARS-CoV-2 RT-PCR Testing on Symptomatic Individuals Attending Healthcare Centers during 2020 in Bahia, Brazil

**DOI:** 10.3390/v14071549

**Published:** 2022-07-15

**Authors:** Felicidade Mota Pereira, Aline Salomão de Araujo, Ana Catarina Martins Reis, Anadilton Santos da Hora, Francesco Pinotti, Robert S. Paton, Camylla Vilas Boas Figueiredo, Caroline Lopes Damasceno, Daiana Carlos dos Santos, Daniele Souza de Santana, Danielle Freitas Sales, Evelyn Ariana Andrade Brandão, Everton da Silva Batista, Fulvia Soares Campos de Sousa, Gabriela Santana Menezes, Jackeline Silveira dos Santos, Jaqueline Gomes Lima, Jean Tadeu Brito, Lenisa Dandara dos Santos, Luciana Reboredo, Maiara Santana Santos, Marcela Kelly Astete Gomez, Marcia Freitas da Cruz, Mariana Rosa Ampuero, Mariele Guerra Lemos da Silva, Mariza S. da Paixão Melo, Marta Ferreira da Silva, Nadja de Jesus Gonçalves dos Santos, Núbia de Souza Pessoa, Ramile Silva de Araujo, Taiane de Macedo Godim, Stephane Fraga de Oliveira Tosta, Vanessa Brandão Nardy, Elaine Cristina Faria, Breno Frederico de Carvalho Dominguez Souza, Jessica Laís Almeida dos Santos, Paul Wikramaratna, Marta Giovanetti, Luiz Carlos Junior Alcântara, José Lourenço, Arabela Leal e Silva de Mello

**Affiliations:** 1Laboratório Central de Saúde Pública Professor Gonçalo Muniz, Salvador 40295-010, Brazil; felicidade.pereira@saude.ba.gov.br (F.M.P.); alinearaujo93@hotmail.com (A.S.d.A.); acmr_bio@hotmail.com (A.C.M.R.); ditodahora@hotmail.com (A.S.d.H.); myllavilas@hotmail.com (C.V.B.F.); carolineldamas@gmail.com (C.L.D.); daianasantos22@hotmail.com (D.C.d.S.); dnssan@hotmail.com (D.S.d.S.); da.ni_freitas@hotmail.com (D.F.S.); evelynaryana@hotmail.com (E.A.A.B.); everton.baptista@gmail.com (E.d.S.B.); fulviasoares@hotmail.com (F.S.C.d.S.); gabrielaas.menezes@gmail.com (G.S.M.); jacky_silveira@hotmail.com (J.S.d.S.); jackgomes_@hotmail.com (J.G.L.); jtbrito@hotmail.com (J.T.B.); dandarassa@hotmail.com (L.D.d.S.); reboredo.oliveira@hotmail.com (L.R.); maiara_santana@hotmail.com (M.S.S.); astetegomezmarcelak@gmail.com (M.K.A.G.); marciafreitascruz@gmail.com (M.F.d.C.); mariana.ampuero@hotmail.com (M.R.A.); marieleguerra@hotmail.com (M.G.L.d.S.); mariza_melo@outlook.com (M.S.d.P.M.); martarego@hotmail.com (M.F.d.S.); nadjajgs@yahoo.com.br (N.d.J.G.d.S.); nubia.pessoa@yahoo.com.br (N.d.S.P.); ramilyaraujo@hotmail.com (R.S.d.A.); taigodim@hotmail.com (T.d.M.G.); vanessanardy@gmail.com (V.B.N.); elaine.gestao@yahoo.com.br (E.C.F.); brenodominguez@gmail.com (B.F.d.C.D.S.); jlasantos17@gmail.com (J.L.A.d.S.); arabelaleal@gmail.com (A.L.e.S.d.M.); 2Department of Zoology, University of Oxford, Oxford OX1 3SZ, UK; francesco.pinotti@zoo.ox.ac.uk (F.P.); robertpaton91@gmail.com (R.S.P.); 3Laboratório de Genética Celular e Molecular, Universidade Federal de Minas Gerais, Belo Horizonte 31270-901, Brazil; sttosta@gmail.com; 4Independent Researcher, Edinburgh EH9 3JT, UK; pwikramaratna@gmail.com; 5Laboratório de Flavivírus, Instituto Oswaldo Cruz Fiocruz, Rio de Janeiro 21045-900, Brazil; giovanetti.marta@gmail.com; 6Department of Science and Technology for Humans and the Environment, University of Campus Bio-Medico di Roma, 00128 Rome, Italy; 7Biosystems and Integrative Sciences Institute, Faculdade de Ciências, 1749-016 Lisboa, Portugal

**Keywords:** SARS-CoV-2, RT-PCR, surveillance, Brazil, testing

## Abstract

RT-PCR testing data provides opportunities to explore regional and individual determinants of test positivity and surveillance infrastructure. Using Generalized Additive Models, we explored 222,515 tests of a random sample of individuals with COVID-19 compatible symptoms in the Brazilian state of Bahia during 2020. We found that age and male gender were the most significant determinants of test positivity. There was evidence of an unequal impact among socio-demographic strata, with higher positivity among those living in areas with low education levels during the first epidemic wave, followed by those living in areas with higher education levels in the second wave. Our estimated probability of testing positive after symptom onset corroborates previous reports that the probability decreases with time, more than halving by about two weeks and converging to zero by three weeks. Test positivity rates generally followed state-level reported cases, and while a single laboratory performed ~90% of tests covering ~99% of the state’s area, test turn-around time generally remained below four days. This testing effort is a testimony to the Bahian surveillance capacity during public health emergencies, as previously witnessed during the recent Zika and Yellow Fever outbreaks.

## 1. Introduction

The coronaviruses HKU1, NL63, OC43, and 229E circulate endemically in the human population, causing first infections in the very young [1] and reinfections throughout life, generally resulting in mild influenza-like illness (ILI) [2]. In contrast, infection with three non-endemic coronaviruses known to infect humans (SARS-CoV-1, MERS-CoV, SARS-CoV-2) more frequently causes severe disease [3,4]. The severe acute respiratory syndrome coronavirus 2 (SARS-CoV-2) was first reported in the Chinese city of Wuhan in December 2019 [5], subsequently causing a pandemic [6,7]. COVID-19, the disease caused by SARS-CoV-2 infection, generally includes fever, fatigue, dry cough, headache, olfactory and taste disorders, diarrhoea, sore throat, and chest pain [3]. The young are typically asymptomatic or present mild disease [8,9]. In cases of severe COVID-19, exacerbated immune responses can lead to respiratory and generalised organ distress [10,11]. The elderly and those with pre-existing diseases or comorbidities are more likely to require hospitalisation, having a greater risk of developing respiratory or multiple organ failure and eventually death [12].

During pandemics, accurate and rapid diagnostic tests are essential to identify localised transmission clusters, measure epidemic size, quantify (a)symptomatic rates of infection, and manage clinical progression, feeding into the planning and implementation of adequate interventions aimed at mitigating public health impacts. Previous infection with SARS-CoV-2 can be identified using serological tests that detect anti-viral immunoglobulins [13]. In contrast, SARS-CoV-2 infected individuals are regularly identified by successful amplification of viral RNA from biological samples using reverse transcriptase polymerase chain reaction (RT-PCR) assays. A myriad of RT-PCR test kits for the COVID-19 diagnosis are now available, targeting a multitude of viral regions such as helicase (Hel), nucleocapsid (N), transmembrane (M), envelope (E), envelope glycoproteins spike (S), hemagglutinin-esterase (HE), open reading frames ORF1a and ORF1b, and RNA-dependent RNA polymerase (RdRp) [13,14,15]. The World Health Organization (WHO) recommends the use of a combination of at least two molecular targets (including at least one conserved region) to prevent cross-reaction with other endemic coronaviruses as well as to detect potential SARS-CoV-2 genetic drift. Early in the pandemic and among possible tests, the E gene assay followed by a confirmatory assay using the RdRp gene represented the “gold standard” utilized for first line screening of suspected COVID-19 cases [16,17,18].

The first confirmed infection by SARS-CoV-2 in Brazil was reported in the city of São Paulo on 26 February 2020 [19]. The application of local non-pharmaceutical interventions (NPIs) has diverged from most strategies in European and North American countries, with no strict lockdown put in place in Brazil [20]. The initial phase of the epidemic in Brazil was fuelled by international travellers, mostly from Italy, China, and France [21]. During the first four months of the epidemic in the largest urban centres of São Paulo and Rio de Janeiro, the effective reproductive potential (Re) of SARS-CoV-2 was reduced by short-term NPIs put in place during March 2020, with Re remaining above 1 after NPI relaxation [22]. In Minas Gerais, the second largest state-population in Brazil, Re was reduced in March 2020, and remained well above 1 shortly afterwards. By early June 2020, serosurveys across the country were measuring large spatial variations in seroprevalence, with the highest seroprevalence in northern cities, in particular within the Amazon state [23,24]. By 30 November 2020, 6,341,965 cases and 173,120 deaths attributed to SARS-CoV-2 had been reported in the country, the largest reported numbers of any country in South America [25]. The epidemic in Brazil during 2020 is likely to have been underreported both in total number of cases and in deaths [26,27,28]. As with many other regions of the world, the direct impact of SARS-CoV-2 infection and indirect impact of the epidemic in Brazil have affected disproportionately different strata of the population, with large differences observed depending on socio-demographic factors such as race and class [29,30,31], regional health infrastructure [32], and between those with and without pre-existing comorbidities or mental health conditions [33,34,35].

The first reported SARS-CoV-2 infection in the state of Bahia was on 6 March 2020 in the second largest city of Feira de Santana, in a 34-year-old woman who was in Italy a few days before the first COVID-19 symptoms appeared [36]. Bahia has an estimated population of >15 million people and is the fifth largest state in Brazil (in area), sharing borders with eight other states. The state has the third largest regional airport system and one of the largest port systems in the country, which makes the flow of people constant and on large scales. Bahia is one of Brazil’s poorest states, facing debilitating socioeconomic inequalities. During the past ten years economic growth has been strong in the state, although concentrated geographically, leaving behind the poorest municipalities, especially those located in semiarid rural areas. The state has the highest illiteracy rate in Brazil (13%), and its capital city (Salvador) has the highest proportion of people residing in subnormal agglomerate sectors (33%) [37]. These factors contribute to a life expectancy which is currently 73.9 years (69.5 for men, 78.7 for women) [37], well below the national average [37].

By 30 April of 2020, the total number of cases in Bahia was 2867, reaching 166,154 by 31 July and exceeding 401,419 by 30 November. Similar to the response at the country level, Bahia did not strictly implement a lockdown. The state did, however, implement several NPIs to hamper the spread of the virus. The first statewide measures were introduced on 16 March, including a mandate for infections confirmed by either a clinic or laboratory to undergo domiciliary quarantine, and a ban on gatherings larger than 50 individuals. A state of emergency was declared on 18 March, when a total of 27 cases had been reported, followed by the first restrictions on long distance movement within Bahia as well as to and from other states. On 9 April, with cases under 1.000, a state of public calamity was declared, leading to activation of further resources to tackle the epidemic. Mandates for mask and hand sanitizing gel in public places and services were issued for the first time on 13 April. A ban on nocturnal travel came into place on 26 June for certain localities, after which localities remained under mobility bans, which continued to be updated with regularity during 2020. Similar to other Brazilian states, the Bahian health authorities set up a real-time reporting system of SARS-CoV-2 infections at the state level during the pandemic, featuring frequent updates in local media outlets and broadcasters. Throughout the pandemic, such data have been invaluable in raising public awareness and complementing genomic surveillance data, with which the state of Bahia has a long and successful history (e.g., in surveillance of dengue virus and during the recent large outbreaks of Zika and Yellow Fever). However, such data lack individual-level metadata, which can contribute to a better understanding of the drivers and characteristics of transmission, dispersion, and disease. In this study, we present and analyse a large dataset of individual-level SARS-CoV-2 RT-PCR tests performed within the state of Bahia between March and November 2020, representative of the period of the first epidemic wave. Tests were performed in symptomatic individuals presenting at public health care centers with influenza-like illness. Contrary to the cohorts of other studies analysing the dynamics of RT-PCR test results, which have been limited to SARS-CoV-2 infected individuals [38,39], this Bahian cohort is a random sample of the population presenting with symptoms compatible with COVID-19, not necessarily those infected with SARS-CoV-2. In this study, we use General Additive Models (GAM) to explore the individual, sample, and test metadata driving cohort positive rates, test result delays, and individual probability of a positive test considering the time from symptom onset.

## 2. Materials and Methods

### 2.1. RT-PCR Full Dataset from Bahia

We looked at 222,515 RT-PCR tests for SARS-CoV-2 managed by the Central Laboratory of Public Health of Bahia (LACEN) and performed within its licensed public laboratories for the time period 28 February to 29 November 2020. The population cohort was composed of individuals presenting to public health care facilities with clinical symptoms compatible with SARS-CoV-2 infection, including at least three of the following: fever, fatigue, dry cough, headache, olfactory and taste disorders, diarrhoea, sore throat, and chest pain.

Each sample (data entry) contained the following variables: date of request, date of sample collection, date of reported symptoms, date of result, individual identifier (anonymised), age, individual gender, nationality, race, municipality of residence, state of residence, sample type, kit test used, gross domestic product per capita of municipality of residence, IDEB (education index) of municipality of residence, population density of municipality of residence, time (days) between collection and symptom onset, days since first test, time (days) for test result, and test result (positive, negative). A summary of the total number of tests dependent on several of the available variables is presented in Appendix A. The state of Bahia is composed of N = 417 municipalities (municípios in Portuguese), which are the smallest geographical unit reported. A summary of tests per municipality is presented in Appendix A. Data for each municipality in Bahia state were collected from the Brazilian Institute of Geography and Statistics (IBGE) (www.ibge.gov.br, accessed on 1 May 2021). This included: population density (habitants per Km^2^), gross domestic product per capita, referred to as GDP (produto interno bruto in Portuguese), and number of education years attended from the final four of the basic system, referred to as IDEB (número de anos finais de ensino fundamental in Portuguese). A summary of the sociodemographic data variables is presented in Appendix A. The catchment areas (regions for which testing was performed) of each laboratory are summarized in Appendix A. For certain analyses, we considered a higher-order spatial dimension based on the official geographical areas, termed meso-regions, of which the borders are described in Appendix A. A total of eight different test kits were reported in the dataset (Appendix A), and nine different public laboratories processed the samples (Appendix A).

### 2.2. RT-PCR Sub Datasets from Bahia

The RT-PCR tests were sorted into three separate sets for the analyses presented in the results section: (i) all tests (N = 222,515); (ii) tests from individuals with >1 tests and at least one positive (up to the first positive test) including all individuals that reported multiple dates of symptoms (N = 220,417); and (iii) samples from patients with >1 tests and at least one positive, including solely patients that reported a single date of symptoms (N = 6560).

### 2.3. SARS-CoV-2 Time Series from Brazil

Data on weekly notified cases of infection with SARS-CoV-2 were made available by the Brazilian Ministry of Health and collated by the COVIDA network at https://github.com/wcota/covid19br, accessed on 7 June 2021. Notified cases are defined as cases confirmed by laboratory diagnostics without full detail on methodology, and their reporting is stated as not being associated with the date of symptom onset (i.e., the time lag between onset, testing, and reporting is unknown). On two occasions in our analyses, we used a Pearson’s test to evaluate (quantify and check statistical significance) the correlation between cases and deaths due to SARS-CoV-2 infection. These tests were performed in R v3.6.3 [40] using the base stats function *cor.test().*

### 2.4. Modelling RT-PCR Results

Generalised Additive Models (GAMs) were implemented in the R programming system using the package mgcv v1.38.1 [41] and R v3.6.3 [40]. Models were compared and selected in a stepwise down procedure from the most complex structure using the Akaike Information Criterion (AIC). Details on tested models, their structure and selection, and code examples are available in Appendix A.

## 3. Results

### 3.1. Reported SARS-CoV-2 Epidemic in Brazil and Bahia

Brazil is composed of 26 states plus a federal district (distrito federal in Portuguese) (Figure 1A). According to the reported numbers by the Brazilian Ministry of Health (BrMoH), between March 2020 and June 2021 the epidemic presented large differences between states both in incidence of cases (Figure 1B,C) and death rates (Figure 1D). Overall, the reported mortality rate in each state varied widely in the first few months of the epidemic, then became reasonably stable for most states around September–October 2020 (Figure 1D). Overall, the temporal progression of the epidemic up to June 2021 in the state of Bahia (BA, Northeast) in terms of incidence of deaths and cases was just under that of the entire country (Figure 1C,D). The reported case data highlighted a clear first wave both at the country and Bahia levels in the winter of 2020 (June–September), with a trough in November and a resurgence in December (Figure 1B). By June 2021, there were large cumulative differences in incidence of both cases and deaths at the state level, although with a reasonable correlation of 0.53 between the two (*p*-value = 0.003, Pearson’s test) (Figure 1E). There were large cumulative differences in incidence of cases and deaths at the municipality level within Bahia as well (Figure 1F). By then, the four largest municipalities (Salvador: population 2.8 M, Feira de Santana: 620 K, Vitória da Conquista: 340 K, Camaçari: 300 K) had an intermediate to high incidence of cases and deaths, while two small municipalities (Itororó: population 20,388 and Itapé: 8526) had the highest incidence of cases and deaths, respectively. Overall, the correlation between cases and deaths at the municipality level was 0.58 (*p*-value < 2.2 × 10^−16^, Pearson’s test), just slightly higher than that observed at the state level.

### 3.2. Dynamics of SARS-CoV-2 RT-PCR Testing in Bahia

We next explored the dynamics of reported RT-PCR testing in Bahia during the time period of the first wave of SARS-CoV-2 (up to November 2020) using the data subset (i) (see RT-PCR sub-datasets for details). Testing was performed by nine laboratories (Appendix A), and while a few laboratories had a regional catchment area, others performed tests across the entire state (Appendix A). Considering the high variation in counts between weekdays and weekends typical of COVID-19 reporting (seen in Figure 1B), we aggregated case and RT-PCR test counts per week. The number of RT-PCR tests per week in Bahia varied significantly between week 9 (end of February) and week 48 (end of November) (Figure 2A). Tests were close to zero up to week 15 (early April), steadily increased afterwards, and reached a maximum of 11,088 on week 35 (late August), five weeks after the peak of reported cases. The positive rate was close to one up to week 15 (early April), likely due to the small number of tests performed on individuals presenting with COVID-19 compatible symptoms (Figure 2B). It then reached a minimum of 0.21 on week 17 (mid-to-late April), after which it presented a fluctuating trend up to week 48 (late November).

With the exception of its proximity to one before week 16, the positive rate reproduced an initial wave and a late resurgence in November (Figure 2B). There appeared to be a time lag between the positive rate and the reported cases during the initial wave. Although there was a marked temporary reduction in testing between weeks 45–47 (early to mid-November, Figure 2A), this did not have an impact on the positive rate (Figure 2B).

To gain insight into the spatial evolution of the positive rate over the weeks, we mapped the 417 municipalities in Bahia at four equally spaced time points, generally representative of four apparent epidemic phases (at the start, during, after, and on resurgence). Early on in the epidemic, in week 18 (late April), few municipalities reported testing (Figure 2C). At week 28 (mid-July) during the first wave, most municipalities in the east coast had positive rates above 0.5, but only a few across the state reached > 0.75 (Figure 2D). After the first wave of reported cases in week 38 (mid-September), the vast majority of municipalities had tests reported, with the east coast now having a lower positive rate than ten weeks before, and some of the municipalities in the interior presenting > 0.75 positive rate (Figure 2E). Later in the year, during the resurgence in reported cases in week 48 (late November), the vast majority of municipalities with test data had a positive rate above 0.5, and a large number of municipalities presented a positive rate > 0.75 (Figure 2F). The resurgence in reported cases in November was therefore being mirrored by an increase in positive rates across the entire state.

The weekly mean number of days between sample collection and test result (test turn-around time) was variable throughout the observation period (Figure 3A), and was generally under four days. Testing was more frequent among females (Figure 3B), those aged 20–60 years (Figure 3C), and those reported with Mixed, Yellow (directly translated to English from the officially category in the data, in Portuguese “Amarela”), or Unknown race (Figure 3D). Testing was more frequently performed by the laboratory Laboratório Central de Saúde Pública Professor Gonçalo Moniz (LCDSPPGM, Figure 3E), and the test kit used varied significantly by week (Figure 3F). Among laboratories, LCDSPPGM performed a total of 201,617 tests in the observation period, approximately ~36 times more than the next laboratory with more tests performed (LMDRRDVDC, total of 5624). The most used test kit was Charité (E) with 87,285 tests, followed by BIOMOL with 67,429 tests. The relative number of samples for each category within the variables gender, age, and race remained relatively stable throughout time (Figure 3B–D). During the observation period, four out of nine kits (Charité, BIOMOL, TaqPath, Allplex) dominated testing, and were sequentially used for varying periods of time (Figure 3F). A total of 143,783 tests were positive and 78,732 were negative for SARS-CoV-2. Overall, 94.8% of unique individuals had a single test performed, 4.5% had two, and 0.68% had at least three. A summary of total tests per available variable is presented in Appendix A.

We restricted the full RT-PCR dataset (Figure 2 and Figure 3 and Appendix A) to subset (ii) starting at week 15, when test numbers started to rise. Data subset (ii) included tests from individuals with >1 tests and at least one positive (up to the first positive test), including all individuals who reported multiple dates of symptoms (see RT-PCR sub-datasets for details). This data restriction was used in order to interpret the positive rate (PR = positive tests/total tests) as a proxy for incidence of new SARS-CoV-2 cases, as including repeated positive tests with different collection dates per individual would introduce a bias towards a measure of prevalence instead. This RT-PCR sub-dataset had 220,417 tests from 209,186 unique individuals. Using GAMs, we fitted the PR versus week of collection. Using the Akaike’s Information Criterion (AIC), model selection was performed over 512 model variants that included the possibility of different combinations of variables and effects between the variables (see Model 1 in Appendix A for details).

To extract a visualisation of the resulting GAM data fitting of the PR in time, it is necessary to perform model predictions by setting all included model variables to particular values. We set variables to their mean if numerical or mode if categorical, and extracted the fits per performing laboratory. We extracted the effects of different variables by calculating the odds ratios (OR) between model predictions over the entire observation period (time-independent) based on specific values of the variable of interest when fixing all other variables to their mean or mode (see Appendix A for details). For presentation of OR, a reference value/category for each variable of interest (i.e., OR = 1) was defined based on the mean/mode of the variable. Considering the state-wide catchment range (~99% of municipalities) and larger number of tests performed (~90% of tests) within the LCDSPPGM laboratory, we focused on this lab and the variables which had statistically significant effects in the main text (Figure 4), with the results of all other laboratories and variables presented in Appendix A.

Overall, the selected model was able to closely approximate the PR of the LCDSPPGM laboratory catchment area over time (Figure 4A). While there were more tests performed for females than males, the gender OR was slightly higher for males at 1.09 (95% CI 1.06–1.13), highlighting a higher probability of a positive test result (Figure 4B). The reported race of the tested individual had little impact on the chances of a test being positive (Figure 4C), while nasopharyngeal secretion samples had a higher chance of being positive, with OR 1.09 (95% CI 1.0–1.21) (Figure 4D). Compared to those aged 50 years (used as reference), individuals aged 5 had a lower chance of a positive test, with an OR of 0.54 (95% CI 0.48–0.64), while those aged 69 had the highest OR at 1.15 (95% CI 1.08–1.21) (Figure 4E). Demographic variables of the municipality of residence influenced the chances of a positive test. GDP (Figure 4F) and density (Figure 4G) had a positive relationship with PR for most of these variables’ range. At high GDP and density, however, this monotonic relationship was replaced by a trough in the OR of having a positive test result. As such, individuals living in municipalities with high (but not the highest) GPD and density had lower chances of having a positive test when presenting with COVID-19 symptoms. A similar relationship was observed on the higher range of IDEB (education), although without a clear monotonic, positive relationship of OR on the lower range of the variable (Figure 4H). The trends of the OR of a positive test for the higher end ranges of socio-demographic variables were not driven by an underlying strong correlation between GDP, IDEB, or density of the municipalities; in fact, only three municipalities had a combination of high GDP, IDEB, and density (Appendix A). Although the catchment areas and total number of tests performed varied significantly across laboratories (Appendix A), the observations described in Figure 4 for the catchment area of the LCDSPPGM laboratory were consistent across laboratories (Appendix A).

We extracted the time-dependent OR of a positive test for each variable (i.e., depending on the week of sample collection by the LCDSPPGM laboratory, Appendix A). For values of the variables gender, sample type, age, and GDP, the OR was stable in time, mimicking the results presented with the time-independent OR in Figure 4. For race, test kit, and density, the OR of different variable values varied significantly with time, with no discernible pattern, which is likely the reason for the wide uncertainty ranges of these variables in the time-independent OR in Figure 4. The most revealing time-dependent pattern was for IDEB; before week 35, the municipalities with the lowest IDEB had the highest OR for a positive test, while after week 35 the OR became highest among municipalities with the highest IDEB. Interestingly, week 35 marks the turning point at which the positive rate began to increase prior to the second wave (Figure 2).

### 3.3. Dynamics of RT-PCR Test Turn-Around Time in Bahia

Using the same data subset (ii) as used for the analyses shown of Figure 4, we used GAMs to fit the test turn-around time (days between sample collection and test result, Figure 3A). Using AIC, we performed model selection for 54 model variants, including the possibility of different combinations of variables and effects between the variables (see Model 2 in Appendix A for details). Considering the predominance of the LCDSPPGM laboratory in the dataset, we focused on this lab and on the variables which had statistically significant effects in the main text (Figure 5), with the results of all other laboratories and variables presented in Appendix A.

The selected model was able to capture the test turn-around time variation by week (Figure 5A). Considering the entire observation period, the (time-independent) OR related to individual age suggested that testing was prioritized towards older ages (shorter turn-around time), with a small overall effect (Figure 5B) that was nonetheless stable with time (Appendix A). The meso-region of a patient’s output suggested small time-independent OR differences (Figure 5C). In contrast to age and meso-region, the laboratory variable had the strongest effects on test turn-around time (Figure 5D). Compared to the LCDSPPGM laboratory, the laboratories LDFEEM (0.74, 95% CI 0.59–0.92), LDIDIDCDS (0.78, 95% CI 0.61–0.92), and LEDAIEV (0.7, 95% CI 0.56–0.85) had OR < 1, having thus performed testing ~20–30% quicker. A few other laboratories had mean OR < 1 or just above 1, although with confidence intervals that largely overlapped with that of LCDSPPGM.

### 3.4. Probability of RT-PCR Positive Test by Time from Symptom Onset

Using data restricted to dataset (iii) (see RT-PCR sub datasets for details) and selecting only individuals who had at least one positive test and with a single reported date of symptoms, we estimated the probability of a positive RT-PCR test dependent on the reported time from symptom onset (Figure 6). We tested 16 models, including the variables of time from symptom onset, age, sample type, test kit, and performing laboratory. The selected model (see Appendix A for details) included only the test kit and laboratory variables as relevant. The probability of having a positive test at 10 days post-symptom onset was estimated to be 0.98 (CI 95% 0.97–0.99), 0.68 (CI 95% 0.56–0.77) at 20 days, and 0.24 (CI 95% 0.07–0.56) at 30 days (Figure 6A). These probabilities were different in two ways when compared to previous estimations based on patient cohorts with longitudinal testing (e.g., [42], presented in Figure 6 in blue and pink depending on sample type). The first difference was that sample type was not selected as a relevant variable for the cohort of this study, likely because of an enrichment towards naso-related samples (62% nasal swabs, 37% nasopharyngeal swabs, totalling 99%). The second difference was that the estimated probabilities were very high, likely because 98.05% of individuals with more than one test stopped testing after a positive result (in contrast with the typical testing in longitudinal cohorts, in which a positive test is followed up by at least one negative test, as in [42]).

We therefore restricted dataset (iii) to the 1.95% of individuals (N = 234 tests) for which any number of negative tests (>0) were obtained after the last positive test. We explored the same model variants as for the results of Figure 6A. The selected model included age, test kit, and laboratory, although none were statistically significant (see Appendix A for details). As such, the difference between the AIC and the model selected using all the data (Figure 6A) was virtually zero. The estimated probability of a positive test from this model more closely approximated previous estimates (Figure 6B). According to the model, the overall probability of having a positive test at 10 days after the reported date of symptom onset was 0.55 (CI 95% 0.37–0.81), and only 0.03 (CI 95% 0.01–0.18) at 20 days. Overall, only 0.2% of all tests (dataset (iii)) were performed after day 25, for which the probability of a positive test would already be close to zero (0.009, CI 95% 0.007–0.19).

## 4. Discussion

With reference to the officially reported number of SARS-CoV-2 infections in the Bahian state of Brazil, we have herein described the epidemic progression in that state during the year of 2020. Using a large number of RT-PCR tests from a random cohort of individuals presenting with COVID-19 compatible symptoms to local public health care facilities, we have described how certain factors associated with individuals and samples dictated the positive rate, test turn-around time, and probability of a positive test considering the time from symptom onset.

We have shown that officially reported cases demonstrate a first epidemic wave between May and October 2020, followed by a second longer wave that was still in effect by June 2021. Up to that month, the state of Bahia had both a cumulative incidence per 100 K individuals and a death rate below that reported for the country as a whole. Compared to the state of Rondônia (Midwest), with the highest number of COVID-19 deaths per 100 K individuals, and the state of Roraima, with the highest number of cases per 100 K individuals, Bahia had ~56% fewer deaths and ~59% fewer cases, respectively. We found that the positive rate of the RT-PCR samples captured the entirety of the first and start of the second waves present in the officially reported cases. There was a time lag during the first epidemic wave, with the RT-PCR positive rate peaking and decreasing earlier than the officially reported cases. Possible explanations for this could include cases being initially underreported at the state level [43,44] and time lags being present in the officially reported cases, which indeed state that reporting timings are not always associated with onset of symptoms or estimated infection date (see SARS-CoV-2 time series from Brazil). In contrast, there appeared to be no time lag between the two variables at the onset of resurgence later in the year. There was a temporary crash in testing between weeks 45–47 that did not influence the positive rate, which could be explained by the occurrence of local elections across all municipalities in the state during those weeks.

The RT-PCR dataset described in this study included nine licensed public laboratories totalling 222,515 tests. The proportion of tests varied according to age, gender, race, laboratory, and test kit. With the exception of the test kit, these proportions were reasonably stable over time. Test kits for infectious diseases (e.g., dengue virus, influenza A) are typically made available in large quantities by the BrMoH. During the 2020 phase of the SARS-CoV-2 pandemic, the BrMoH focused distribution on the Brazilian BIOMOL kit (BioManguinhos and Oswaldo Cruz Foundation). However, each laboratory under the centralized coordination of Central Laboratory of Public Health of Bahia (LACEN) remained responsible for managing necessary extra stock, regularly resorting to private shipping of alternative test kits when test demand was high. This resulted in the dominant test kit changing regularly over time, with at least ten kit switches up to November 2020. Although we aimed to explore whether the test kit used influenced the probability of a sample being positive, this irregular relationship of the dominant test kit with time is likely to have been the reason that, in most of our analyses, the kit test was not found to significantly explain any particular patterns in the observed data.

We were unable to explore the specificity of SARS-CoV-2 clinical triage and its relationship with test results. This was because the data did not include records of symptoms per sample. While the majority of tests were positive, suggesting that triage for testing based on symptoms was efficient to a certain degree, the remaining ~36% of tests were negative for SARS-CoV-2. A possible future research path would be to explore those clinical records in order to assess the specificity of clinical triage for SARS-CoV-2 testing in Bahia. Lack of symptom records and no multiple testing per sample restricted our ability to consider the possibility of false positive or false negative tests. It is difficult to envisage how false test results could influence the results herein described, however, we argue that the large number of samples included in each of the presented analyses should have hampered such effects.

Among the participating laboratories, the Laboratório central de saúde pública professor gonçalo moniz (LCDSPPGM), located in the capital city of Salvador, performed ~90% of all tests (a total of 201,617), and it had the largest test catchment area, including ~99% of Bahian municipalities. Considering its data representativeness, we focused on this laboratory when presenting results related to fitting the positive rate and time turn-around time of the RT-PCR samples in time. The results demonstrated that among those presenting with compatible symptoms, males were more likely to be infected with SARS-CoV-2. Indeed, the association of male gender with a more pronounced COVID-19 clinical outcome has previously been described [29,45,46]. For this data cohort, it was not possible to discern whether other factors could have determined the observed link, e.g., differences in healthcare access between genders. In Brazil, women may access healthcare more frequently than men [47,48]. As demonstrated by a higher number of RT-PCR tests in females in our dataset, this could have resulted in more females attending healthcare upon the onset of any COVID-19 compatible symptoms, artificially boosting the likelihood of negative tests for SARS-CoV-2.

Age had the strongest effect on the positive rate, with a 5-year-old having half the odds ratio of having a positive SARS-CoV-2 test of a 69-year-old individual. Although unclear why, this enrichment in positive test results with age among those experiencing COVID-19 compatible symptoms has been reported in other Brazilian states [49,50]. While older individuals are known to be more susceptible to both SARS-CoV-2 infection and COVID-19 symptoms [12], in order to explain the result it is necessary to assume that sampled symptomatic younger individuals were carriers of unknown respiratory pathogens with similar clinical outcomes, which would be circulating during the autumn and winter months of the study period (e.g., influenza, respiratory syncytial viruses).

We found that, during the first epidemic wave, the chances of a positive test were higher among those living in municipalities with the lowest education index (IDEB). During the SARS-CoV-2 resurgence later in the year this relationship was reversed, and the chances of a positive test became higher among those living in municipalities with a higher IDEB. The first wave may have thus had an unequal impact among socio-demographic strata in Bahia, as has been reported for other regions of Brazil [51,52]. In estimations of the determinants of test turn-around time (time between sample and test result), the performing laboratory was the most important factor. Several laboratories (LDFEEM, LDIDIDCDS, LEDAIEV) performed ~20–30% quicker than LCDSPPGM. However, LCDSPPGM tested approximately 159, 221, and 46 times more tests than these laboratories, respectively, highlighting the major effort and efficiency of testing in that laboratory.

Finally, although we could not explore the impact of a few important factors, such as symptom variation (e.g., mild versus severe), we corroborated previous estimates that the probability of testing positive for SARS-CoV-2 generally drops with time from symptom onset, more than halving by about two weeks and converging to zero by three weeks. There were differences between our estimations and others in the literature. The largest differences were found when the analyses were performed using the entire dataset, and likely arose from an existing enrichment in individuals who stopped testing after their first positive test, something that is not present in typical literature datasets and analyses, which tend to be based on patient cohorts with longitudinal repeated testing between the first positive test and later multiple negative tests. We thus restricted our dataset to include solely individuals for whom we could find N > 0 negative tests after the last positive test. This restriction resulted in a cutoff of 98.8% of available data (N = 234). The low number of tests and individuals used, together with a possible effect of false test results (which we could not address in our analyses), were likely the cause of the remaining small differences in the probability of testing positive after symptom onset that we report versus other estimations already present in the literature. These differences include a slightly higher probability of a positive test closer to symptom onset and a sharper decrease in that probability with time from symptom onset.

## 5. Conclusions

By exploring a vast number of SARS-CoV-2 RT-PCR tests we have described and helped to explain a number of relevant data signatures, such as the association of positive tests with gender, age, and socio-demographic factors, as well as the dependency of the probability of a positive test with time from symptom onset. In this manuscript, we commit to making this unique dataset publicly available to the research community, which we hope can inform future SARS-CoV-2 research. As during other recent public health emergencies in Brazil (e.g., Zika, yellow fever), this and other studies are a testimony that investment by Bahian authorities has built local surveillance capacity, as well as that a continuing investment is necessary in order to guarantee and maintain the critical generation of data that can help researchers and policy makers better tackle emerging pathogens in the state of Bahia.

## Figures and Tables

**Figure 1 viruses-14-01549-f001:**
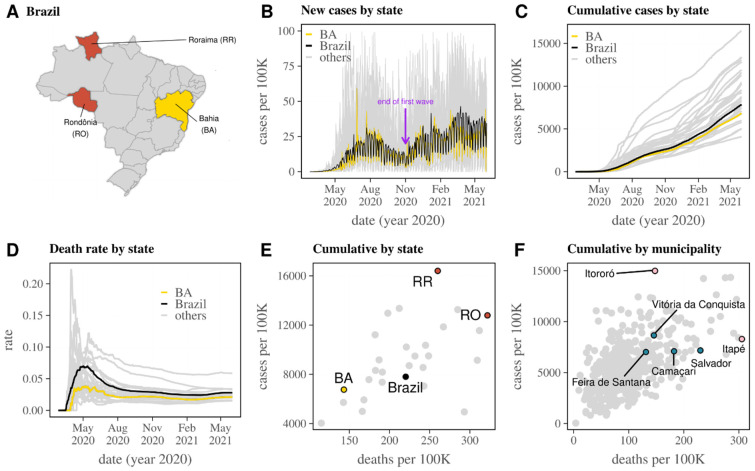
Summary of reported SARS-CoV-2 epidemic in Brazil and Bahia. (**A**) Map of Brazilian states (top), with those relevant for the other panels highlighted with different colors. (**B**) Reported cases per day per state. The purple arrow marks the start of November 2020, when the first trough marks the end of the initial wave of cases. (**C**) Cumulative case incidence (per 100 k residents) per state. (**D**) Death rate (cumulative cases/deaths) per state. (**E**) Cumulative incidence of cases versus deaths per state up to the end of November 2020. (**F**) Cumulative incidence of cases versus deaths in Bahia municipality up to the end of November 2020. (**A**–**E**) Bahia state (BA) in yellow; Brazil (BR) average in black; Roraima state (RR) and Rondônia (RO) in red; Bahia’s largest four urban centers/municipalities in blue; Bahia municipalities with highest incidences in pink; all other data in grey (states and Bahia municipalities).

**Figure 2 viruses-14-01549-f002:**
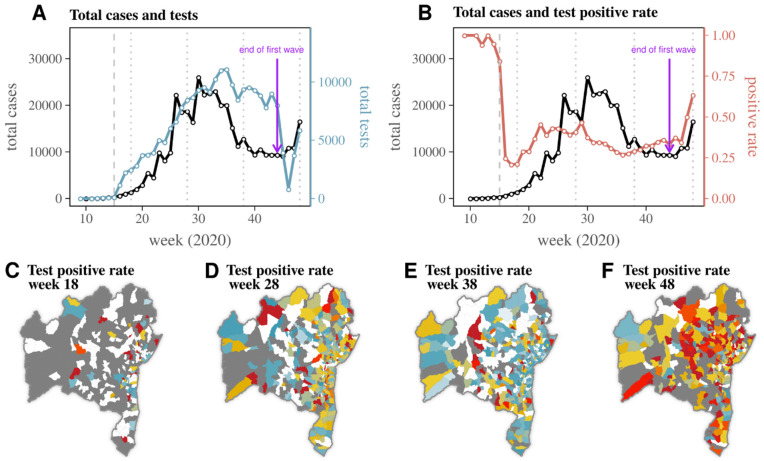
Reported SARS-CoV-2 cases and RT-PCR testing in Bahia during the first wave. (**A**) Total cases (black) and RT-PCR tests (blue) per week in Bahia. (**B**) Total cases (black) and positive rate (red) per week in Bahia. (**A**,**B**) Purple arrow marks the first week of November 2020 when daily incidence (Figure 1B) presents a trough indicative of the end of the first epidemic wave in the state. The dashed grey line marks week 15, before which the positive rate is artificially high due to low number of tests performed. (**C**–**F**) Mapping of positive rate (PR) per municipality in Bahia for (**C**) week 18, (**D**) week 28, (**E**) week 38, and (**F**) week 48. These weeks are marked in panels (**A**,**B**) as vertical dotted lines. Municipalities with no tests are coloured in grey, while those with tests are coloured according to scale on the far right.

**Figure 3 viruses-14-01549-f003:**
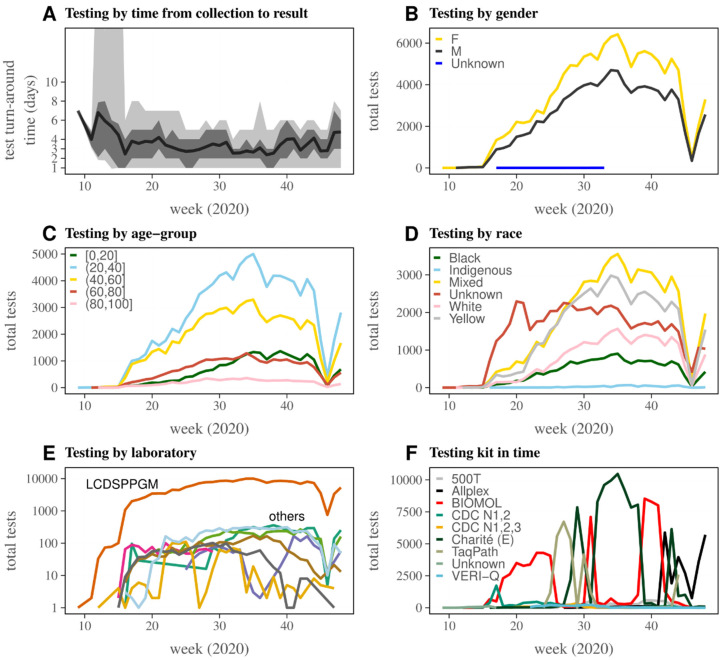
Summary of SARS-CoV-2 RT-PCR testing in Bahia, March-November 2020. (**A**) Test turn-around time (number of days between sample collection test result). Dark grey shaded area is the 95% quantile of the test turn-around time across all tests, light grey shaded area is the 75% quantile, and the black line is the mean; all data are per week. (**B**) Total weekly tests by gender and (**C**) age group in years (trimmed to maximum 100 years), both according to color legend. (**D**) Total weekly tests by reported race according to color legend. (**E**) Total weekly tests by performing laboratory, with highest performing laboratory LCDSPPGM in purple and others with varying colors. For a list of all laboratories, see Appendix A. (**F**) Total weekly tests by test kit used according to color legend. For details on kits, see Appendix A. (**B**–**F**) For total of tests per variable, see Appendix A.

**Figure 4 viruses-14-01549-f004:**
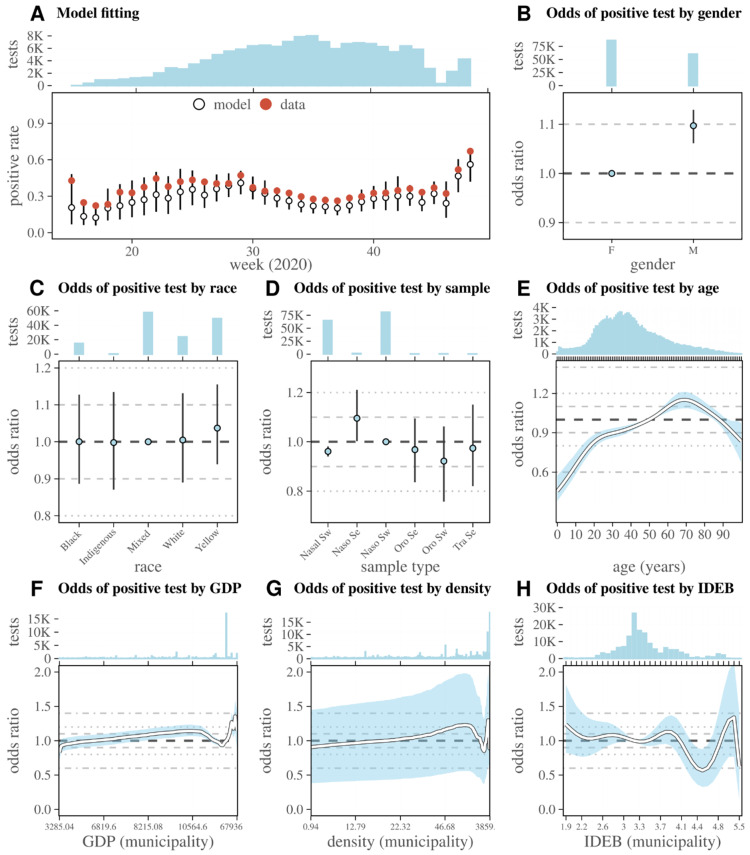
SARS-CoV-2 positive rate and odds ratio for a positive test dependent on other variables (laboratory LCDSPPGM). (**A**) Model fit to positive rate per week, with model in white full circles and data in red full circles. (**B**–**H**) Odds ratio of a positive test depending on (**B**) gender, (**C**) race, (**D**) sample type, (**E**) age, (**F**) GDP, (**G**) density, and (**F**) IDEB of the patient’s municipality. (**A**–**H**) Top subpanel presents the total tests per variable value/category on the x-axis (the K in the y-axis refers to thousands). Values of reference (odds = 1) are the mode of each variable except for age, for which the reference was 50 years. Points (discrete variables) and lines (continuous variables) are the mean odds, and the whiskers and areas are the 95% percentile. (**D**) Sample type key: “Naso Sw” = nasopharyngeal swab, “Oro Se” = oropharyngeal secretion, “Naso Se” = nasopharyngeal secretion, “Oro Sw” = oropharyngeal swab, “Tra Se” = tracheal secretion, “Nasal Sw” = nasal swab. The presented fit was extracted from the GAM model by setting the variable laboratory to LCDSPPGM; for other laboratories, see Appendix A. Variables presented are the statistically significant ones (see Model 1 in Appendix A for details).

**Figure 5 viruses-14-01549-f005:**
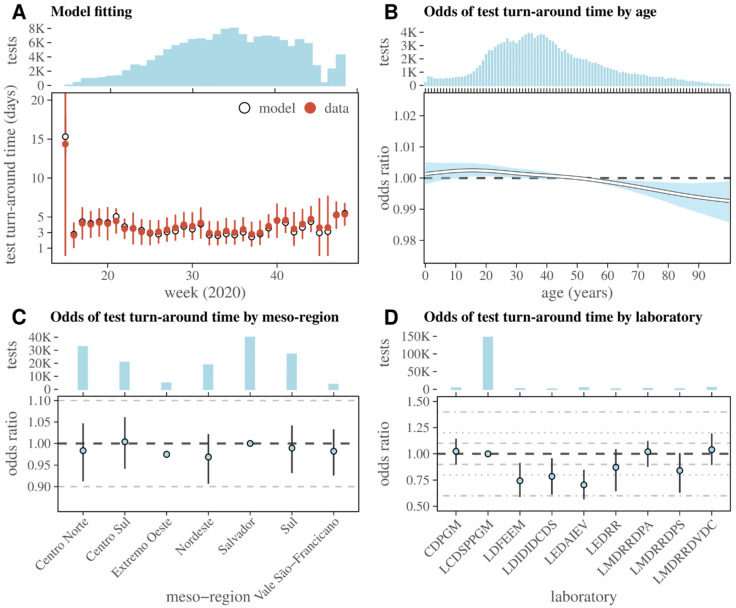
SARS-CoV-2 test turn-around time and odds ratio for time taken dependent on other variables (laboratory LCDSPPGM). (**A**) Model fit to test turn-around time (days between sample collection and test result) per week. Odds ratio of a positive test depending on (**B**) age, (**C**) meso-region of the tested individual, and (**D**) performing laboratory. (**A**–**D**) Top subpanels present the total tests (blue bars). The K in the y-axis refers to thousands. Whiskers are the 95% percentile and full circles the mean of the odds ratio. Values of reference (odds = 1) are the mode of each variable. The presented fit was extracted from the GAM model by setting the variable laboratory to LCDSPPGM; other laboratories are shown in Appendix A.

**Figure 6 viruses-14-01549-f006:**
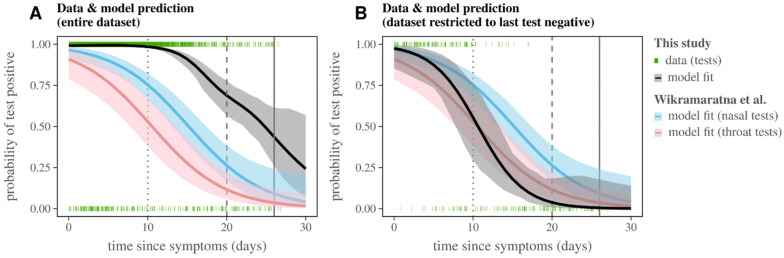
SARS-CoV-2 probability of positive test dependent on time from symptom onset. (**A**) Model fit (black line and grey area) for positive and negative tests (red bars at 1 and 0, respectively) in this study’s dataset (iii) (see RT-PCR sub-datasets for details), selecting only individuals who had at least one positive test and with a single reported date of symptom onset. (**B**) Model fit (black line and grey area) for positive and negative tests (red bars at 1 and 0, respectively) in this study’s dataset (iii) (see RT-PCR sub datasets for details), restricted to those for whom the last test was negative. (**A**,**B**) The pink and blue lines (and respective areas) are the predictions by Wikramaratna et al. [42] dependent on sample type, with nasal samples in blue and throat samples in pink.

## Data Availability

For computational code, we used free-to-use existing R tools (see Methods), and we provide example code in Appendix A. All used sub-datasets have been made available in a FigShare repository entitled “RT-PCR SARS-CoV-2 datasets, Bahia, Brazil, 2020” under doi:10.6084/m9.figshare.c.5823812.v1.

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
