# Peer review of "Dynamics and Determinants of SARS-CoV-2 RT-PCR Testing on Symptomatic Individuals Attending Healthcare Centers during 2020 in Bahia, Brazil"

_viruses, 2022, doi:10.3390/v14071549_

Round 1

Reviewer 1 Report

The topic is very interesting and the methods that the authors used are appropriate. There is enough justification about the research and the results are clear. The presentation is well and organization of the paper is good. I have no major comments in this step.

Author Response

We thank the reviewer for the positive comments and the time spent reviewing the original version of the manuscript. All the best.

Reviewer 2 Report

In “Dynamics and determinants of SARS-CoV-2 RT-PCR testing on 2 symptomatic individuals attending healthcare centers during 2020 in Bahia, Brazil” Pereira and colleagues present and analysed a large dataset of SARS-CoV-2 RT-PCR tests performed within the state of Bahia between March and November 2020, representative of the period of the first epidemic wave. They use General Additive Models (GAM) to explore the individual, sample and test meta-data driving cohort positive rates, test result delays and individual probability of positive test given time since symptoms. The manuscript contain important data from the SARS-COV2 first wave in their territory and deserve being registered in a publication. However, the connection between each part of the paper should be improved:

a) The main objetive and the main finding is not fully clear in the abstract.

b) The first five paragraphs of the introduction describes data and information form the pandemics in Brazil, but information about the study can be found mainly in the last paragraph. A paragraph explaining the reader why the present study is important could be included in the introduction.

c) Method section is too small relative to the other sections of the manuscript. An increased method section could be good for the paper readability and compreession.There are some sentences found in the results section that can be migrated to the method section (lines 294- 306). Additionally, person’s test was used to investigate the correlation between some variables but the methods did not describe that this test will be used.

d) Results section are sound. However, an increase in the  space between each graphics in the figures could facilitate its comprehension (y-axis title could be more distant form the graphic on its left - Fig 1,2,3,4,5). The inclusion of the y-axis title in the top subpanel which depict the total tests per variable in figure 4 and 5 will also increase the comprehension of the graphics.

e) Finally, each result included in the result section should be discussed (e.g. one of the main finding: dependency of the probability of a positive test with time since the symptoms could not be found in discusion.

F) if appropriated, please consider change the expression “test delay” to “test turn-around time”.
